biomimetics/fluid mechanics

leading-edge vortex, swift, delta, swept-back wings, particle image velocimetry

**Author for correspondence:**
Erin E. Hackett
e-mail: ehackett@coastal.edu

# Leading-edge vortices over swept-back wings with varying sweep geometries

William B. Lambert[1], Mathew J. Stanek[2], Roi Gurka[2] and Erin E. Hackett[2]

[1]Department of Math, Computer Science, and Physics, Roanoke College, Salem, VA, USA
[2]Department of Coastal and Marine Systems Science, Coastal Carolina University, Conway, SC, USA

RG, 0000-0002-8907-6663; EEH, 0000-0002-1294-6501

Micro air vehicles are used in a myriad of applications, such as transportation and surveying. Their performance can be improved through the study of wing designs and lift generation techniques including leading-edge vortices (LEVs). Observation of natural fliers, e.g. birds and bats, has shown that LEVs are a major contributor to lift during flapping flight, and the common swift (*Apus apus*) has been observed to generate LEVs during gliding flight. We hypothesize that nonlinear swept-back wings generate a vortex in the leading-edge region, which can augment the lift in a similar manner to linear swept-back wings (i.e. delta wing) during gliding flight. Particle image velocimetry experiments were performed in a water flume to compare flow over two wing geometries: one with a nonlinear sweep (swift-like wing) and one with a linear sweep (delta wing). Experiments were performed at three spanwise planes and three angles of attack at a chord-based Reynolds number of 26 000. Streamlines, vorticity, swirling strength, and *Q*-criterion were used to identify LEVs. The results show similar LEV characteristics for delta and swift-like wing geometries. These similarities suggest that sweep geometries other than a linear sweep (i.e. delta wing) are capable of creating LEVs during gliding flight.

## 1. Introduction

With an increased use of micro air vehicles (MAVs) new aerodynamic challenges arise [1]. The tasks of MAVs are many and include activities that require long-distance flight, large payloads and tight manoeuvres. Aerodynamic forces play a major role for a given task with respect to flight performance. The goal is to increase aerodynamic efficiency, which typically involves maximizing lift while minimizing drag. One approach to increasing lift through wing design involves the generation of a

leading-edge vortex (LEV) that reduces flow separation at the leading edge of the wing by forming a large region of circulation on the upper surface.

The LEV's stability and ability to augment lift depend on numerous variables, such as wing shape, sweep angle, and angle of attack [2]. LEVs are formed at sharp leading edges and cause flow reattachment to the wing after separation [3]. The fluid over the leading edge accelerates downward resulting in enhanced lift generation called nonlinear or vortex lift [4]. In addition to the extra lift generated, the LEV also aids in the distribution of force across the chord of the wing and delays flow separation [4]. A recent review by Eldredge & Jones [5] outlined the physical mechanisms of LEV formation and its role in generating additional lift over the upper surface of certain wing geometries. However, this lift comes at a cost and it is noteworthy that LEVs also increase drag at all non-zero angles of attack [6].

Over the past 70 years, the phenomenon of stationary LEVs on low aspect ratio (AR) wings has been studied extensively through analytical efforts [4,7,8], experiments [4,9,10] and numerical simulations [11–13]. The most notable example of this type of wing is a delta wing, where LEV formation has been extensively studied in a variety of different settings. Taylor $et\ al.$ [14] tested model delta wings with sweep angle $\lambda = 50°$ at a variety of attack angles ($\alpha$) and chord-based Reynolds numbers ($Re_c$). They observed LEVs for $2.5° \le \alpha \le 15°$ at $Re_c = 13\,000$. The LEVs observed showed a dual nature across a range of Reynolds numbers $8700 \le Re_c \le 34\,700$ at $\alpha = 7.5°$. Gordnier & Visbal [15] performed computational simulations of a similar delta wing with $\lambda = 50°$. Simulations of attack angles $5° \le \alpha \le 15°$ at $Re_c = 26\,000$ showed a dual LEV system at $\alpha = 5°$ with the system gradually diminishing for each subsequent increase in attack angle. At higher $\alpha$, a vortex breakdown was observed with an increase in the unsteadiness of the flow. Jin-Jun & Wang [16] studied delta wings over a range of sweep angles, $45° \le \lambda \le 65°$, attack angles, $0° \le \alpha \le 32°$, for $Re_c = 6000$, $12\,000$ and $18\,000$. It was observed for $Re_c = 12\,000$ and $18\,000$ that larger sweep angles would result in vortex breakdown at higher angles of attack.

Although LEVs have been used for enhancing lift for low AR wings, LEVs for lift augmentation also appear to exist on higher AR wings of animals [17–21]. Studies of animal flight have reported that LEVs, characterized as transverse spanwise vortices, can form above wings in two modes: unsteady and steady. Unsteady corresponds to an LEV that is continuously generated during flapping motion and attaches to the wing surface, grows in size and eventually detaches and sheds. Steady refers to an LEV that remains stationary above the wing, similar to the phenomenon found on delta wings, which is the focus of this study. This form of LEV can be found both on owls' wings [22] and on swept-back wings of gliding swifts [23]. Owls use stationary LEVs for producing high lift when silently approaching prey, when their wings maintain a high angle of attack [22]. The common swift (*Apus apus*), which stays aloft for over 10 months at a time and has recently broken the record for the longest sustained flight for birds, demonstrates itself as a very efficient flier [24]. The formation of LEVs over the swift's wings occurs during gliding modes as well as during flapping flight.

The wing of a swift consists of two parts, the short arm wing close to the body and the longer hand wing further out [23]. The geometries of each part of the wing are unique and aid in the production of the aforementioned LEV as a means to enhance lift. The arm wing is more rounded on the leading edge relying on traditional forms of lift, whereas the hand wing is highly swept back and has a sharp leading edge thereby producing an LEV. Swift wings are characterized by a nonlinear sweep-back angle, which differs from delta wings that have a linear sweep-back angle. Therefore, one can assume that similar to natural fliers, geometrical features of the wings can be used as a passive control mechanism that may enhance lift by altering the flow over them. Muir $et\ al.$ [25] compared the flow over delta wings with different trailing edges, where one was inspired by the swift. Measurements of the flow in the spanwise-normal plane using particle image velocimetry (PIV) for various angles of attack and $Re_c$ show that LEVs were produced. They observed flow structures comprising dual and triple LEVs developing over both wing configurations and concluded that LEV formation is independent of the trailing-edge geometry, which supported earlier observations by Taylor $et\ al.$ [14].

The present study is an investigation of LEV formation comparing vortex development over a nonlinear swept-back wing (swift) to that over a linear swept-back wing (delta). The flows over these wings are measured using PIV in an open water channel facility at three angles of attack at $Re_c = 26\,000$. Measured water velocities are analysed to evaluate LEV formation and results for the different wing geometries are compared.

## 2. Experiments

Flows over two different wing geometries were measured using PIV in a recirculating water flume. The two geometries consisted of a delta and swift-like wing. Experiments were performed at three different

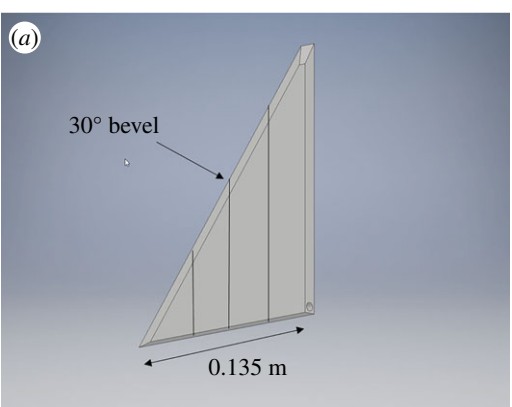
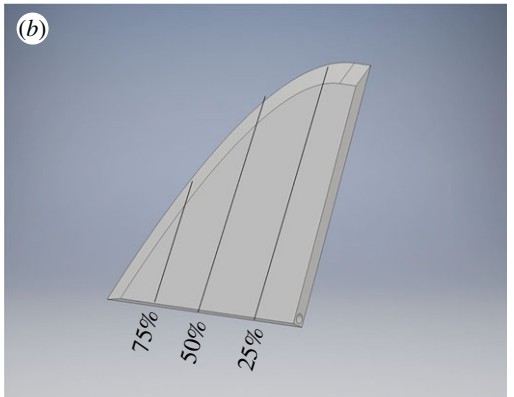

**Figure 1.** Three-dimensional CAD renderings of the delta (*a*) and swift-like (*b*) wings. These wing models were created using a Mark Forged 3D printer with onyx fibre. Black lines on wings show the PIV cross sections measured at 25, 50 and 75% of the span.

attack angles, $\alpha = 0°, 20°, 30°$, which were set using a digital level with an accuracy $\pm 0.1°$ and re-checked/measured following each experiment to ensure that the wing angle did not change during data collection. Flow measurements were performed in the streamwise and wall-normal plane at three different spanwise positions over the wing, $z = 0.041$ m, $0.073$ m and $0.104$ m, where $z = 0$ is defined as being the root chord position. These spanwise locations correspond to approximately 25%, 50% and 75% of the span of the wing from the root, referred herein as the quarter, half and three-quarter plane, respectively. For each experiment, 500 PIV image pairs were acquired. All experiments were performed at a chord-based Reynolds number of $Re_c = U_0 c / v = 26\,000$ where $U_0$ is the streamwise uniform flow velocity (estimated from the flow rate), $c$ is the root chord length (0.15 m), and $v$ is the kinematic viscosity of water ($1.11 \times 10^{-6}$ m$^2$ s$^{-1}$ at 22°C).

The wing designs shown in figure 1 were three-dimensional printed (Mark Forged) using onyx fibre. A delta wing is used because previous studies have shown development of LEVs over them at high angles of attack with limited stall [25,26]; while the swift design was inspired by the geometrical form of the *Apus apus* wing [23]. The traditional delta wing has dimensions of 0.15 m $\times$ 0.125 m $\times$ 0.007 m with an AR of 3.3 and a linear sweep of $\lambda = 50°$. The swift-like wing has dimensions of 0.16 m $\times$ 0.135 m $\times$ 0.007 m with an AR of 3.4 and a nonlinear sweep-back angle. The position of the leading edge for the swift wing is defined as $10.33z^3 + 7.158z^2 + 0.049z - 0.016$. Each wing has an additional 0.01 m in the spanwise direction located at the root to connect it to the mounting hardware, and a bevel of 30° from the horizontal on the leading edge to create a sharp leading edge [25]. Note that all of the wings have the same trailing-edge geometry enabling us to isolate the role of the leading edge when comparing the experimental results.

Experiments were conducted in a recirculating water flume located in the Environmental Fluids Laboratory at Coastal Carolina University. The water flume is composed of a 15 m long open channel with a cross section of 0.5 m $\times$ 0.7 m, where the middle 5 m is made of transparent glass. At either end of the channel are two large holding reservoirs. The water depth in the channel was held constant at 0.36 m. A large centrifugal pump with a flow rate capacity up to 3028 LPM (800 GPM) is used to generate flow and the pump rotation speed is controlled via a variable frequency drive. The uniform flow velocity within the channel was estimated as $U_0 = Q/A = 0.19$ ms$^{-1}$, where $Q$ is the flow rate measured by a rotary flow meter and $A$ is the wetted cross-sectional area of the water in the flume. The flow velocity yields a $Re_c = 26\,000$ for all experiments.

The flow was measured using PIV [27] and was performed in the streamwise and wall-normal plane. The experimental set-up is depicted in figure 2, where the wing was placed upside down in the channel and the flow over the wing was imaged below the wing by illuminating a plane of light from beneath the channel. A dual pulsed Nd:YAG laser (Quantel Inc. EverGreen 145) operating with 145 mJ per pulse and wavelength of 532 nm was used to illuminate the flow. The light sheet was approximately 1 mm thick and the particles were fused borosilicate glass microspheres of 11 µm average diameter (Potters Industries). A 29MP double exposure CCD camera (PowerView$^{TM}$) with dynamic range of 12 bits operating at 1 Hz with a Nikon 200 mm lens and 20 mm extension tube imaged a planar area of 0.31 m $\times$ 0.21 m. A total of 500 image pairs were collected per experiment. Each image pair was analysed using cross-correlation (Insight 4G$^{TM}$, TSI Inc.) to estimate two velocity components in the streamwise (*u*) and wall-normal (*v*) directions over the two-dimensional plane (*x* and *y*). Each image

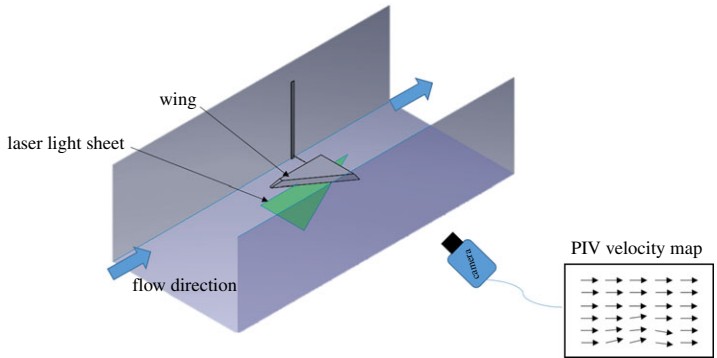

**Figure 2.** Schematic of the PIV set-up within the water flume. The wing section was placed upside down in the flume and the light sheet was produced underneath the flume. A camera outside the flume captured image pairs used to generate velocity vector maps.

pair was divided into interrogation regions of 64 pixel × 64 pixel with 50% overlap for the cross-correlation, and vector maps were subsequently filtered for outliers with both local and global filters. Uncertainty on the instantaneous velocities is estimated as 2–4% of the measured value while an uncertainty of 8–10% is estimated for the computed instantaneous velocity gradients.

## 3. Vortex identification methods

Two vortex identification methods, along with other fluid characteristics, are applied to identify vortex locations in the flow: swirling strength and $Q$-criterion. Chong *et al.* [28] define swirling strength as an area where rotation dominates rate-of-strain. This scenario results in complex eigenvalues for the rate-of-deformation tensor. Maps of the velocity gradients are computed using a least-squares differencing technique to generate a two-dimensional velocity gradient tensor at each location for each map. Swirling strength is calculated as the magnitude of the imaginary eigenvalue of these velocity gradient tensors, if one existed. The existence of complex eigenvalues suggests the presence of a vortex. This calculation results in a series of swirling strength maps, which is then ensemble averaged over all maps.

The $Q$-criterion defines a vortex based on areas where the magnitude of the vorticity tensor is greater than the magnitude of the rate-of-strain tensor [29]. The anti-symmetric part ($[\Omega] = \frac{1}{2}(\nabla\mathbf{v} - (\nabla\mathbf{v})^{\mathrm{T}})$) and symmetric part ($[S] = \frac{1}{2}(\nabla\mathbf{v} + (\nabla\mathbf{v})^{\mathrm{T}})$) of the velocity gradient tensor are computed for every position within each velocity map, where the velocity gradients are again calculated using least-squares differencing. A tensor here is denoted with bold face font type and square brackets. $Q$ is then computed as $\frac{1}{2}[\|[\Omega]^2\| - \|[S]^2\|]$, where the double vertical bars denote a Euclidean norm. $Q > 0$ indicates the presence of a vortex. This calculation results in a series of $Q$ maps that is subsequently ensemble averaged over all maps.

## 4. Results

The flow over the wings in all configurations is inherently turbulent and these analyses of the resulting LEVs represent ensemble averages of the vortex characteristics. There are many different methods to determine whether or not an area of interest is considered a vortex; thus, four main criteria were used to assess whether or not a particular velocity field had an LEV. We examined the streamlines of the mean velocities for areas of circulation, and identified locations of high vorticity, swirling strength, and positive $Q$-criterion. Each of these are evaluated on coherency and location with respect to the wing. Because vortex identification criteria are debatable, somewhat subjective, and prone to error, the use of multiple criteria helps to avoid associated biases. For example, the use of vorticity alone can be misleading because any shear in the flow also produces large vorticity. Also, the application of $Q$-criterion and swirling strength based on two-dimensional measurements of a three-dimensional flow can yield misleading results. Most differences between vortex identification methods applied in two-dimensional boundary layer flow are associated with the sensitivity of the methods to small-scale vortices [30]. Lastly, it should be noted that the measurements only provide a projection of any LEV onto the measurement plane and the magnitude of the projection varies with the wing geometry along the wingspan making it challenging to compare between planes (along the span). Results of

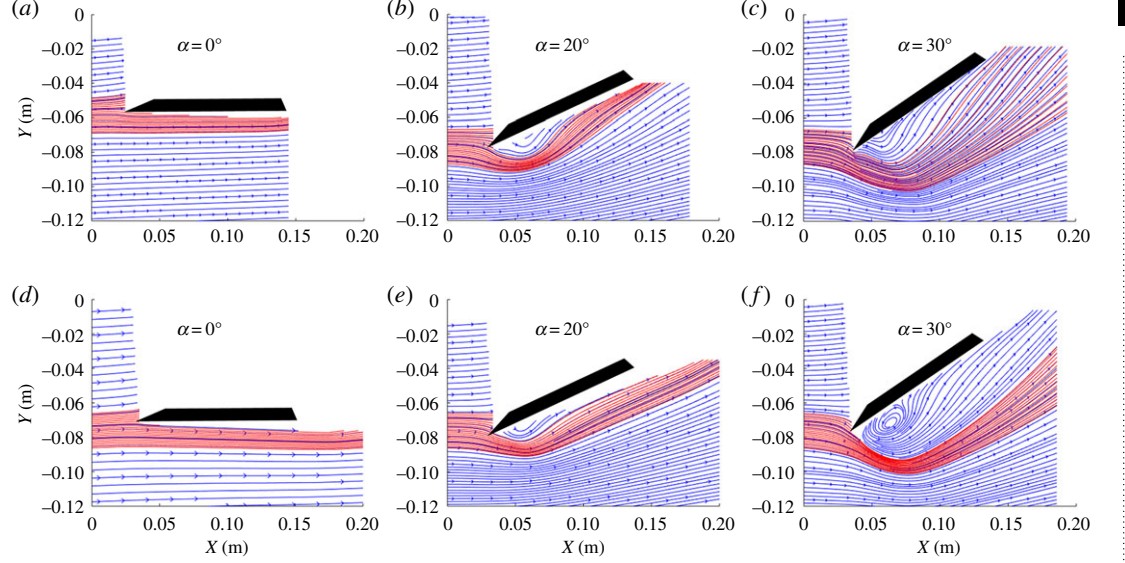

**Figure 3.** Streamlines of the mean velocity field at the quarter plane for a delta wing at $\alpha = 0°$ (a), $\alpha = 20°$ (b) and $\alpha = 30°$ (c) and swift wing at $\alpha = 0°$ (d), $\alpha = 20°$ (e) and $\alpha = 30°$ (f). The red lines highlight streamlines that intersect with the leading edge of the wing.

using the various analysis techniques are described below, where collectively they suggest that an LEV was present for both the delta and swift-like wings for attack angles of $\alpha = 20°$ and $\alpha = 30°$.

## 4.1. Streamlines

We evaluate flow separation over the upper wing surface by observing streamlines of the fluid flow in a stationary frame of reference. The streamlines incident at the leading edge of the wing are examined for reattachment, and when there is no reattachment of these streamlines we presume that flow separation occurred. A streamline in the two-dimensional plane is defined by $dy/dx = v/u$. Streamlines of the flow are computed for the mean velocity maps. Figure 3 depicts streamlines over both the delta and swift wings at the quarter plane. The streamlines depict flow reattachment patterns occurring at angles of attack varying from $0°$ to $30°$. The red lines highlight streamlines that intersect with the leading edge of the wing. At $\alpha = 0°$ no separation is observed. Flow reattachment is observed for both wing geometries at $\alpha = 20°$ with a degradation occurring for both at $\alpha = 30°$. This similarity coupled with the similar formation of an area of circulation at the leading edge is indicative of a comparable vortex system forming.

## 4.2. Spanwise vorticity

Given that the measurements were performed using two-dimensional PIV, the spanwise vorticity component, $\omega_z = \partial v/\partial x - \partial u/\partial y$, is estimated from the instantaneous velocity map series. The spanwise vorticity series is ensemble averaged over all maps providing a single map of $\bar{\omega}_z$, where the overbar denotes an ensemble mean. Figure 4 compiles the mean spanwise vorticity distribution for both the delta and swift wing at the quarter plane using the same range of angles of attack as in figure 3. One can observe analogous vorticity contours forming over each wing geometry. The vorticity regions show similar characteristic shapes and magnitudes, and vary in a similar way with changes in the angle of attack. At $\alpha = 0°$, mean vorticity is negligible relative to the cases with higher angles of attack. At $\alpha = 20°$ and $\alpha = 30°$, a concentrated region of spanwise vorticity can be observed for both wing configurations in the leading-edge region—the same region where fluid circulation was observed in figure 3. The coincidence of these regions provides more evidence of an LEV.

## 4.3. Vortex identification

In addition to the streamlines and vorticity occurring simultaneously, swirling strength and Q-criterion also appear in the same region for both the delta (figure 5) and swift (figure 6) wing geometries. Both

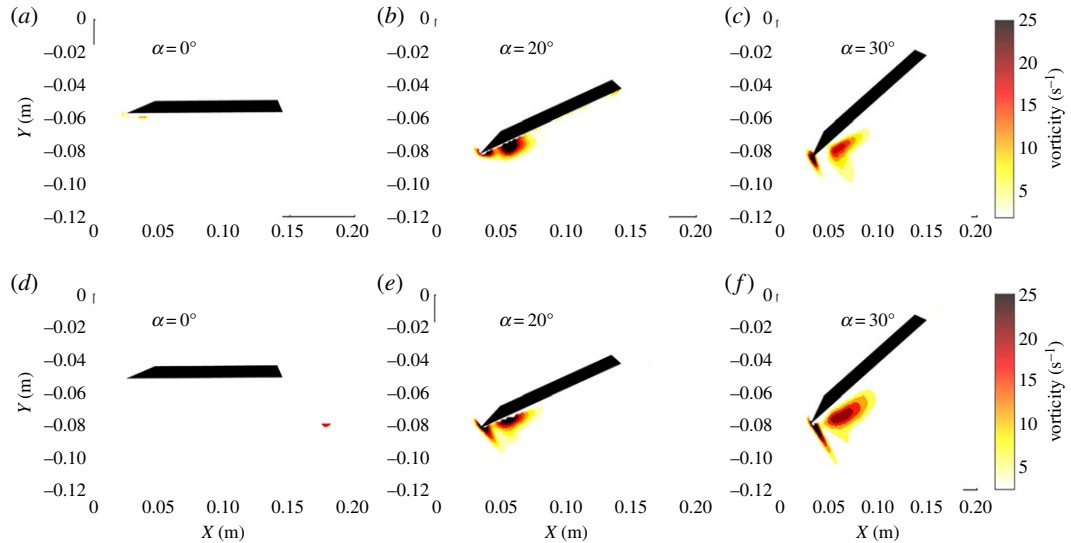

**Figure 4.** Mean vorticity contours for the quarter plane of the delta wing at $\alpha = 0°$ (a), $\alpha = 20°$ (b) and $\alpha = 30°$ (c) and swift wing at $\alpha = 0°$ (d), $\alpha = 20°$ (e) and $\alpha = 30°$ (f).

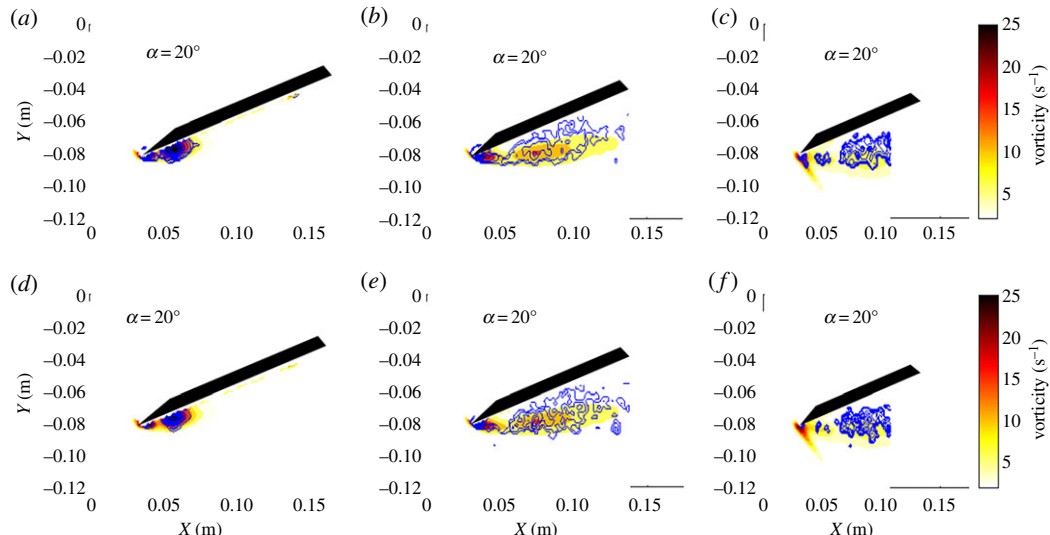

**Figure 5.** Mean vorticity contours for a delta wing at $\alpha = 20°$ with contours of mean swirling strength (a–c) and mean Q-criterion (d–f) overlaid. Each column depicts a different spanwise plane with the first column showing results for the quarter plane (a,d), the second column showing half plane results (b,e), and the third column showing results for the three-quarter plane (c,f).

figures comprise mean vorticity contours with contours of mean swirling strength (a–c) and mean Q-criterion (d–f) overlaid. Each column depicts results for different spanwise planes: quarter plane (a,d), half plane (b,e) and three-quarter plane (c,f). The results presented are for $\alpha = 20°$. The two different techniques present similar results in terms of location and distribution. Both the linear and the nonlinear swept-back wings show that the areas of high circulation previously discussed also contain regions of high mean swirling strength and Q. Obtaining the same flow structure using two different identification techniques compounds the notion that this area of shear or two-dimensional rotation also features the existence of an LEV. It is noteworthy to observe the evolution of the flow structure along the spanwise direction. At the quarter plane, the LEV is small and concentrated at the leading-edge region. At the half plane, the LEV increases in size for both wing geometries and stretches out towards the trailing edge (i.e. in the streamwise direction) as well as increasing its diameter, yielding an oval shape projection of the LEV in the measured plane. At the three-quarter plane, the LEV keeps growing in size but its intensity (based on the mean vorticity) continues to decrease.

Collectively, these observations suggest that swept leading-edge geometries, linear or nonlinear, sustain an LEV over this region once it forms. The flow, passing over the wing, for swept wings

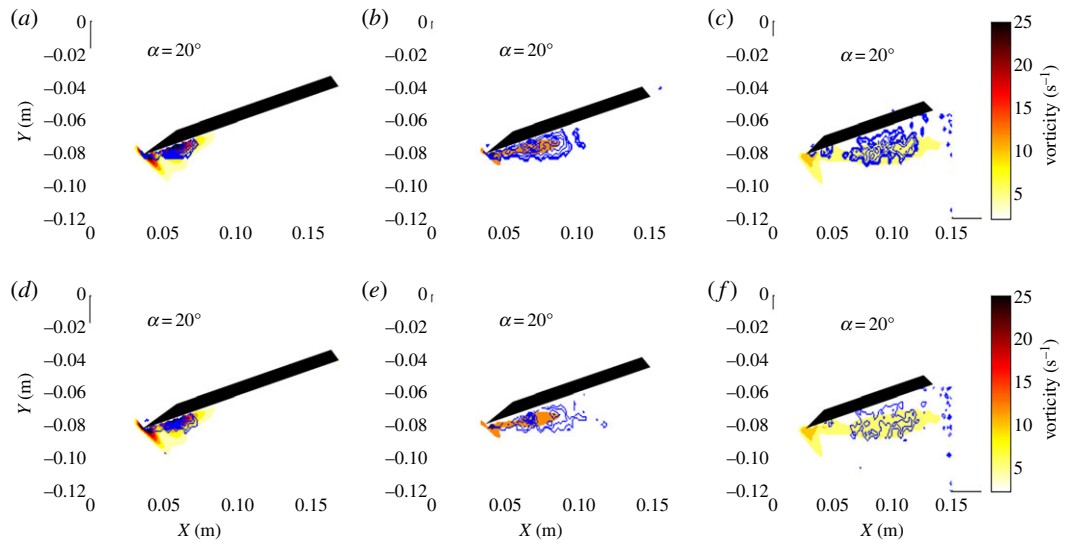

**Figure 6.** Mean vorticity contours for a swift wing at $\alpha = 20°$ with contours of mean swirling strength $(a–c)$ and mean $Q$-criterion $(d–f)$ overlaid. Each column depicts a different spanwise plane with the first column showing results for the quarter plane $(a,d)$, the second column showing half plane results $(b,e)$ and the third column showing results for the three-quarter plane $(c,f)$.

experiences changes in momentum in the spanwise direction (towards the tip) as noted by Muir *et al.* [25]. The presence of a boundary layer (formed over the upper surface of the wing) yields transport of angular momentum in the spanwise direction. Approaching the tip region, the LEV is larger in size but weaker in strength, yet, the wing chord also decreases in this region, which prevents flow reattachment. The vortex at this region would likely be unstable and break down without reattachment of the flow. In the plane closest to the tip, the LEV on the swift-like wing appears more coherent than the delta and may be due to the longer chord available for reattachment on the swift-like wing.

## 4.4. Vertical momentum and circulation

Here we examine whether the observed circulation patterns can be associated with increased lift. In this case, with the wing upside down, vertical momentum associated with lift would be in the negative $y$-direction (i.e. the direction normal to the surface). To unambiguously identify negative vertical velocity, we consider vertical displacements (from the PIV measurements) of at least 1.5 pixels in the downward direction as indicative of negative velocities (i.e. $v < -0.03 \text{ m s}^{-1}$). At each position along the chord, the mean of these negative vertical velocities over the region beneath the wing was computed and squared to estimate the vertical momentum distribution over the chord per unit span $(\rho v^2)$, where $\rho = 1000 \text{ kg m}^{-3}$, and is shown in figure 7 for various planes and angles of attack. This distribution integrated over the chord is proportional to the vertical forces per unit span acting on the wing and is used to evaluate the similarity of the LEV as a mechanism contributing to (total) lift generation for the delta and swift wings. In figure 7, the wing leading and trailing edges are shown as thin vertical lines. For both the delta and swift-like wings, there is no apparent vertical momentum at $0°$ angle of attack, while at higher attack angles ($\alpha = 20°$ and $\alpha = 30°$), we observe vertical momentum near the leading edge that overlaps with the LEV regions previously identified. Similar trends between the two geometries are also observed as the angle of attack and plane change, such as peak vertical momentum location relative to the wing and characteristic widening, which is approximately linear with the increase in angle of attack. Clearly, both wing geometries produce comparable vertical momentum profiles; thus, one can suggest that this lift-generating mechanism is also analogous.

The circulation of the flow associated with the LEV is estimated as $\Gamma = \iint_R \omega_z dA$, where the region, $R$, covered by the LEV is defined as the area below the wing that yields swirling strengths greater than $2 \text{ s}^{-1}$. The circulation is another contributor to the overall lift. The circulations for each spanwise plane and angle of attack for both wings are shown in table 1. The circulation peaks at the mid-span of the wing for both sweep geometries at $\alpha = 20°$, and also for the delta wing at $\alpha = 30°$. The circulations also increase with the angle of attack suggesting more lift is needed at larger angles. The trends of circulation for the delta and swift-like wing geometries were similar supporting

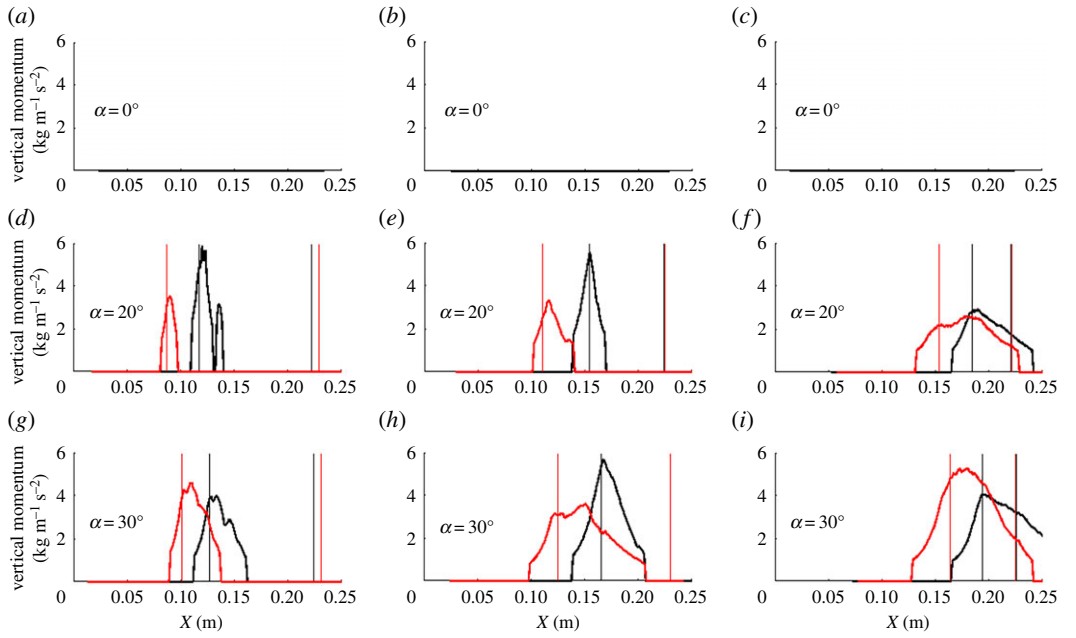

**Figure 7.** Vertical momentum distribution over the chord per unit span for the delta wing (black) and for the swift-based wing (red) for each angle of attack (rows) and each plane (columns). (a–c) Results for $\alpha = 0°$, (d–f) results for $\alpha = 20°$ and (g–i) results for $\alpha = 30°$. Results for the quarter plane are shown in (a,d,g), for the half plane in (b,e,h), and for the three-quarter plane in (c,f,i). The respective vertical lines denote the leading and trailing edge of each wing in each plane.

**Table 1.** Circulation for each angle of attack and spanwise measurement plane for the swift-style and delta wing geometries. Circulations are given in $m^2\,s^{-1}$.

| angle of attack | spanwise plane (%) | circulation: swift | circulation: delta |
|---|---|---|---|
| 0° | 25 | $0.00 \times 10^{-3}$ | $0.00 \times 10^{-3}$ |
| | 50 | $0.00 \times 10^{-3}$ | $0.00 \times 10^{-3}$ |
| | 75 | $0.00 \times 10^{-3}$ | $0.63 \times 10^{-3}$ |
| 20° | 25 | $4.23 \times 10^{-3}$ | $5.92 \times 10^{-3}$ |
| | 50 | $6.45 \times 10^{-3}$ | $10.5 \times 10^{-3}$ |
| | 75 | $4.72 \times 10^{-3}$ | $4.45 \times 10^{-3}$ |
| 30° | 25 | $10.9 \times 10^{-3}$ | $5.88 \times 10^{-3}$ |
| | 50 | $8.73 \times 10^{-3}$ | $10.9 \times 10^{-3}$ |
| | 75 | $7.85 \times 10^{-3}$ | $5.73 \times 10^{-3}$ |

similar LEV characteristics. Collectively, the vertical momentum and circulation, which are contributors to lift, support that there is similar LEV formation for both linear and nonlinear swept-back leading-edge geometries.

## 5. Conclusion

Flows over two different leading-edge wing geometries were measured using PIV in order to further understand the formation of LEVs on swept-back wings. Two swept-back leading-edge geometries were examined: delta (linear sweep) and swift (nonlinear sweep). Experiments at $Re_c = 26\,000$ were conducted at three different angles of attack and flow over the wing was measured in three planes along the spanwise direction to view a representative portion of the vortex formation. The LEVs are identified using several criteria including centralized circulation of streamlines, high spanwise vorticity, and positive vortex identification using swirling strength and Q-criterion. Vertical momentum along the

wing chord was also examined as an indicator of transport, and circulation was estimated for each wing geometry at each spanwise location and angle of attack. No wing geometry exhibited an LEV at $\alpha = 0°$. LEVs are observed for both the delta and the swift at $\alpha = 20°$ and $30°$.

Overall, there appears to be a high level of similarity in the identified LEVs for delta and swift wings. They both show similar development of the LEV along the spanwise direction from the root towards the tip at several attack angles. They also show similarity in the movement of the vortex system centre towards the trailing edge at increasing distances from the root along the wing span, and similar distribution of vertical momentum along the chord. Because an LEV is present with both sweep geometries, there is support for the idea that there may be several leading-edge geometries for swept wings that aid in the creation of an LEV, which enhances lift, as has been shown for the delta wing. Other variables such as trailing-edge geometries, not varied in this study, may also affect the formation of LEVs, as studied by Muir *et al.* [25]. The chord lengths at different spanwise positions differed for the swift and delta wing with the swift having a longer chord near the tip that may allow for reattachment of the flow even at positions close to the tip. This discrepancy may explain the slightly differing results for the swift and delta at the three-quarter plane. In all, these results reveal no major distinctions between the linear and nonlinear swept-back wings on LEV formation, indicating that nonlinear swept-back wings in addition to linear swept-back wings are capable of forming and maintaining a steady LEV.

Data accessibility. Data are available from the Dryad Digital Repository at: https://doi.org/10.5061/dryad.b7g95d2 [31].
Authors' contributions. Conceptualization, E.E.H. and R.G.; methodology, E.E.H., R.G. and W.B.L.; experiment execution, W.B.L., M.J.S. and E.E.H.; data analysis, W.B.L. and E.E.H.; writing, W.B.L., E.E.H., R.G. and M.J.S.
Competing interests. The authors declare no competing interests.
Funding. This research was supported in part by the Research Experiences for Undergraduates programme at the National Science Foundation under award number AGS-1560210.
Acknowledgements. The authors thank Ian Matsko for three-dimensional printing of the wings and Hadar Ben-Gida for providing a model of the swift wing geometry upon which the three-dimensional print model was based.

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
