## [Reviewer comments · Royal Society Open Science]

Review History

RSOS-190514.R0 (Original submission)

Review form: Reviewer 1

Is the manuscript scientifically sound in its present form?

Yes

Are the interpretations and conclusions justified by the results?

Yes

Is the language acceptable?

Yes

Is it clear how to access all supporting data?

Yes

Do you have any ethical concerns with this paper?

No

Have you any concerns about statistical analyses in this paper?

No

Recommendation?

Accept with minor revision (please list in comments)

Comments to the Author(s)

The study is a comparative set of experiments between a linear delta wing and a nonlinear wing shape inspired by a swift. The experiments appear to be of high quality and properly performed, and the article is well-written. The findings of this study are expected to be a valuable contribution to the overall understanding of unsteady flight aerodynamics. However, some recommendations and questions have been identified during the review of the manuscript. Therefore, major revisions are recommended prior to recommending this manuscript for publication.

General comments

=====

For two-dimensional data the various vortex identification methods are essentially equivalent to determine the presence of a vortex (Jeong and Hussain, 1995). It is recommended to provide the rationale for reporting the results of two alternative vortex identification methods, and what difference between the Q-criterion and the swirling strength might physically imply. There is commentary of why vorticity and Q-criterion/swirling strength should both be assessed (Line 178), but clarification on the distinct interpretations of the Q-criterion and swirling strength would be useful.

There is discussion in the Introduction of how some fliers have unsteady or steady leading edge vortices. However, all of the results in this study seem to be based on averaged values. It is recommended to add some details about the unsteady characteristics of the flow. For example, does the reattachment point fluctuate? Is there any transition to turbulence in the separated shear layer as the angle of attack increases? Do these characteristics change for the swift-like wing?

Although no surface pressure and no force measurements were performed, it is recommended to consider either or both of the following points to help complete the description of total lift:

- (i) Is it possible to extract estimates of the near-surface pressure distribution from the PIV data? This information along with the integration of the vertical momentum transfer would help to complete the description of the total lift (assuming that viscous stresses are small).
- (ii) In the absence of pressure field calculations, it could alternatively be valuable to estimate circulation of the identified vortices to give a quantitative description of vortex strength. This calculation would also have obvious implications for total lift.

Minor comments

=====

Line 251: It is unclear how the vertical momentum distribution is calculated. What distance from the surface is the velocity sampled? Please clarify.

Line 253: The velocity-squared momentum transfer term is not necessarily proportional to the total lift. It is directly proportional to the lift due to momentum transfer, but the total lift depends strongly on the surface pressure distribution. Information on the pressure distribution would be required to determine the contribution of momentum transfer to the total lift. It is recommended to limit the generalization to the total lift in the absence of pressure data.

Figures 4, 5 and 6: The gradient in the colormap appears to give only three steps in the calculated

vorticity contours. It is recommended to increase the resolution in the colormap to give a better picture of the variation in vorticity.

Reference 4: Palhamus should be Polhamus

Review form: Reviewer 2

Is the manuscript scientifically sound in its present form?

Yes

Are the interpretations and conclusions justified by the results?

Yes

Is the language acceptable?

Yes

Is it clear how to access all supporting data?

No

Do you have any ethical concerns with this paper?

No

Have you any concerns about statistical analyses in this paper?

No

Recommendation?

Reject

Comments to the Author(s)

Your effort is undermined somewhat by a general understanding that swift wings and an array of wing shapes from other animals, all of which have curved leading edges, all generate LEV's. It would more substantially advance our understanding of the effects of leading edge geometry upon LEV formation if your experimental design would have tested a broader range of variation in curvature until significant differences were observed between curved and straight leading edges. That written, your methods and analysis appear robust for the specific case you tested.

I encourage you to recast your presentation more clearly as a test of a hypothesis than as a descriptive comparison. This applies to the abstract (lines 16-17) and later in the introduction and conclusions.

Lines 22-23 (and conclusions). Your major conclusion is relatively weak, as we already understand that a variety of animal wings can generate LEV's.

Line 31: Delete either "aerodynamic forces such as" or "lift and drag" as what other aerodynamic forces are there under consideration?

Line 34: LEV's generate greater pressure differentials on the wings and whether this is considered lift or drag is depended upon the angle of attack of the wing. My point is that LEV's are not a "free ride" in that they increase lift but also drag, and this inherent cost is worth including in the introduction.

Line 64: This sentence should be recast in terms of advance ratio. When the wings are moving faster than the forward speed of the animal, unsteady aerodynamic effects may dominate. The flapping motions may be just as steady (or unsteady) in slow flight as during cruising flight.

Line 78: Define "strong" in quantitative terms.

Results: The beginning of the results (lines 166-179) is a restatement of the introduction and methods. This should be deleted as it is redundant. Several subsections of the results begin with a restatement of the introduction or methods (or in a couple of cases, conclusions). This includes lines 189-193, 202-207, and 246-255. These cases should be deleted as they are redundant, or components of them should be moved to the introduction or methods.

Figure 1: Consider adding lines to indicate the span-wise locations at which you sampled flow using PIV

Figure 2: This figure isn't needed, as your setup is standard.

Figures 7 and 8: Consider combining these, using two different line colors or solid versus dashed. This would facilitate comparison of the two sets of results.

Decision letter (RSOS-190514.R0)

10-May-2019

Dear Dr Hackett,

The editors assigned to your paper ("Leading-edge Vortices over Swept-back Wings with Varying Sweep Geometries") have now received comments from reviewers. We would like you to revise your paper in accordance with the referee and Associate Editor suggestions which can be found below (not including confidential reports to the Editor). Please note this decision does not guarantee eventual acceptance.

Please submit a copy of your revised paper before 02-Jun-2019. Please note that the revision deadline will expire at 00.00am on this date. If we do not hear from you within this time then it will be assumed that the paper has been withdrawn. In exceptional circumstances, extensions may be possible if agreed with the Editorial Office in advance. We do not allow multiple rounds of revision so we urge you to make every effort to fully address all of the comments at this stage. If deemed necessary by the Editors, your manuscript will be sent back to one or more of the original reviewers for assessment. If the original reviewers are not available, we may invite new reviewers.

When submitting your revised manuscript, you must respond to the comments made by the referees and upload a file "Response to Referees" in "Section 6 - File Upload". Please use this to

document how you have responded to the comments, and the adjustments you have made. In order to expedite the processing of the revised manuscript, please be as specific as possible in your response.

- Data accessibility

If you wish to submit your supporting data or code to Dryad (<http://datadryad.org/>), or modify your current submission to dryad, please use the following link:
<http://datadryad.org/submit?journalID=RSOS&manu=RSOS-190514>

- Competing interests

- Authors' contributions

- Acknowledgements

- Funding statement

on behalf of Professor Brooke Flammang (Associate Editor) and R. Kerry Rowe (Subject Editor)
 openscience@royalsociety.org

Comments to Author:

Reviewers' Comments to Author:

Reviewer: 1

Comments to the Author(s)

The study is a comparative set of experiments between a linear delta wing and a nonlinear wing shape inspired by a swift. The experiments appear to be of high quality and properly performed, and the article is well-written. The findings of this study are expected to be a valuable contribution to the overall understanding of unsteady flight aerodynamics. However, some recommendations and questions have been identified during the review of the manuscript. Therefore, major revisions are recommended prior to recommending this manuscript for publication.

General comments

=====

For two-dimensional data the various vortex identification methods are essentially equivalent to determine the presence of a vortex (Jeong and Hussain, 1995). It is recommended to provide the rationale for reporting the results of two alternative vortex identification methods, and what difference between the Q-criterion and the swirling strength might physically imply. There is commentary of why vorticity and Q-criterion/swirling strength should both be assessed (Line 178), but clarification on the distinct interpretations of the Q-criterion and swirling strength would be useful.

There is discussion in the Introduction of how some fliers have unsteady or steady leading edge vortices. However, all of the results in this study seem to be based on averaged values. It is recommended to add some details about the unsteady characteristics of the flow. For example, does the reattachment point fluctuate? Is there any transition to turbulence in the separated shear layer as the angle of attack increases? Do these characteristics change for the swift-like wing?

Although no surface pressure and no force measurements were performed, it is recommended to consider either or both of the following points to help complete the description of total lift:

- (i) Is it possible to extract estimates of the near-surface pressure distribution from the PIV data? This information along with the integration of the vertical momentum transfer would help to complete the description of the total lift (assuming that viscous stresses are small).
- (ii) In the absence of pressure field calculations, it could alternatively be valuable to estimate circulation of the identified vortices to give a quantitative description of vortex strength. This calculation would also have obvious implications for total lift.

Minor comments

=====

Line 251: It is unclear how the vertical momentum distribution is calculated. What distance from the surface is the velocity sampled? Please clarify.

Line 253: The velocity-squared momentum transfer term is not necessarily proportional to the total lift. It is directly proportional to the lift due to momentum transfer, but the total lift depends strongly on the surface pressure distribution. Information on the pressure distribution would be required to determine the contribution of momentum transfer to the total lift. It is recommended to limit the generalization to the total lift in the absence of pressure data.

Figures 4, 5 and 6: The gradient in the colormap appears to give only three steps in the calculated vorticity contours. It is recommended to increase the resolution in the colormap to give a better picture of the variation in vorticity.

Reference 4: Palhamus should be Polhamus

Reviewer: 2

Comments to the Author(s)

Your effort is undermined somewhat by a general understanding that swift wings and an array of wing shapes from other animals, all of which have curved leading edges, all generate LEV's. It would more substantially advance our understanding of the effects of leading edge geometry upon LEV formation if your experimental design would have tested a broader range of variation in curvature until significant differences were observed between curved and straight leading edges. That written, your methods and analysis appear robust for the specific case you tested.

I encourage you to recast your presentation more clearly as a test of a hypothesis than as a descriptive comparison. This applies to the abstract (lines 16-17) and later in the introduction and conclusions.

Lines 22-23 (and conclusions). Your major conclusion is relatively weak, as we already understand that a variety of animal wings can generate LEV's.

Line 31: Delete either "aerodynamic forces such as" or "lift and drag" as what other aerodynamic forces are there under consideration?

Line 34: LEV's generate greater pressure differentials on the wings and whether this is considered lift or drag is depended upon the angle of attack of the wing. My point is that LEV's are not a "free ride" in that they increase lift but also drag, and this inherent cost is worth including in the introduction.

Line 64: This sentence should be recast in terms of advance ratio. When the wings are moving

faster than the forward speed of the animal, unsteady aerodynamic effects may dominate. The flapping motions may be just as steady (or unsteady) in slow flight as during cruising flight.

Line 78: Define “strong” in quantitative terms.

Results: The beginning of the results (lines 166-179) is a restatement of the introduction and methods. This should be deleted as it is redundant. Several subsections of the results begin with a restatement of the introduction or methods (or in a couple of cases, conclusions). This includes lines 189-193, 202-207, and 246-255. These cases should be deleted as they are redundant, or components of them should be moved to the introduction or methods.

Figure 1: Consider adding lines to indicate the span-wise locations at which you sampled flow using PIV

Figure 2: This figure isn’t needed, as your setup is standard.

Figures 7 and 8: Consider combining these, using two different line colors or solid versus dashed. This would facilitate comparison of the two sets of results.

Author's Response to Decision Letter for (RSOS-190514.R0)

See Appendix A.

Decision letter (RSOS-190514.R1)

03-Jun-2019

Dear Dr Hackett,

I am pleased to inform you that your manuscript entitled "Leading-edge Vortices over Swept-back Wings with Varying Sweep Geometries" is now accepted for publication in Royal Society Open Science.

on behalf of Professor Brooke Flammang (Associate Editor) and R. Kerry Rowe (Subject Editor)
openscience@royalsociety.org

Appendix A

Response to Reviewer 1

Our point-by-point responses are included below where manuscript changes are denoted in italic and are referred to by line number of the revised manuscript.

The study is a comparative set of experiments between a linear delta wing and a nonlinear wing shape inspired by a swift. The experiments appear to be of high quality and properly performed, and the article is well-written. The findings of this study are expected to be a valuable contribution to the overall understanding of unsteady flight aerodynamics. However, some recommendations and questions have been identified during the review of the manuscript. Therefore, major revisions are recommended prior to recommending this manuscript for publication.

Thank-you for your helpful comments and suggestions, which have certainly improved the manuscript.

General comments

1. For two-dimensional data the various vortex identification methods are essentially equivalent to determine the presence of a vortex (Jeong and Hussain, 1995). It is recommended to provide the rationale for reporting the results of two alternative vortex identification methods, and what difference between the Q-criterion and the swirling strength might physically imply. There is commentary of why vorticity and Q-criterion/swirling strength should both be assessed (Line 178), but clarification on the distinct interpretations of the Q-criterion and swirling strength would be useful.

Thanks. There are several existing studies that examine this issue, so we have added a comment and a reference for more information comparing between the swirling strength and Q-criterion vortex identification methods.

L176-178: "Most differences between vortex identification methods applied in 2D boundary layer flow are associated with the sensitivity of the methods to small-scale vortices³⁰."

References:

"30. Chen, Q., Zhong Q, Qi M, Wang X. 2015. Comparison of Vortex Identification Criteria for Planar Velocity Fields in Wall Turbulence, Physics of Fluids, 27, 085101. doi:/10.1063/1.4927647."

2. There is discussion in the Introduction of how some fliers have unsteady or steady leading edge vortices. However, all of the results in this study seem to be based on averaged values. It is recommended to add some details about the unsteady characteristics of the flow. For example, does the reattachment point fluctuate? Is there any transition to turbulence in the separated shear layer as the angle of attack increases? Do these characteristics change for the swift-like wing?

Thank-you. We agree that the term steady vs. unsteady may seem a bit misleading as the flow at these Reynolds numbers is inherently unsteady. The term 'steady' refers to an

average LEV. Indeed, the reattachment point fluctuates; that is partly the reasoning for using multiple identification schemes to quantify the location of the LEV in a statistical manner. We assume that at these Reynolds numbers, the flow is turbulent, and there were no transitional effects. Our general conclusion from this study is that the flow over non-linear swept-back wings at this aspect ratio behave similarly to a linear one. Consequently, our study is focused on steady LEVs (gliding mode of flight). We agree this was not necessarily clearly communicated in the introduction during the discussion of unsteady versus steady LEVs. We have clarified this in the revised manuscript in both the introduction and results sections.

L66-68: *“Steady refers to an LEV that remains stationary above the wing, similar to the phenomenon found on delta wings, which is the focus of this study. This form of LEV can be found both on owls’ wings²² and on swept-back wings of gliding swifts²³.”*

L166-170: *“The flow over the wings in all configurations is inherently turbulent and these analyses of the resulting LEVs represent ensemble averages of the vortex characteristics. There are many different methods to determine whether or not an area of interest is considered a vortex; thus, four main criteria were used to assess whether or not a particular velocity field had a leading-edge vortex.”*

3. Although no surface pressure and no force measurements were performed, it is recommended to consider either or both of the following points to help complete the description of total lift:
 - (i) Is it possible to extract estimates of the near-surface pressure distribution from the PIV data? This information along with the integration of the vertical momentum transfer would help to complete the description of the total lift (assuming that viscous stresses are small).
 - (ii) In the absence of pressure field calculations, it could alternatively be valuable to estimate circulation of the identified vortices to give a quantitative description of vortex strength. This calculation would also have obvious implications for total lift.

We appreciate the suggestion. We cannot estimate the pressure over the surface in our experimental setup because our upper flow boundary is a free surface. Following your second suggestion, we have estimated the circulation over the wing at the leading edge region, and summarized the results in a table added to the revised manuscript. The results are consistent with the vertical momentum results and the general conclusions of the study.

L262-271: *“The circulation of the flow associated with the LEV is estimated as $\Gamma = \oint_R \omega_z dA$, where the region, R , covered by the LEV is defined as the area below the wing that yields swirling strengths greater than 2. The circulation is another contributor to the overall lift. The circulation for each spanwise plane and angle of attack for both wings are shown in Table 1. The circulation peaks at the mid-span of the wing for both sweep geometries at $\alpha=20^\circ$, and also for the delta wing at $\alpha=30^\circ$. The circulations also increase with the angle of attack suggesting more lift is needed at larger angles. The trends of circulation for the delta and swift-like wing geometries were similar supporting similar LEV characteristics. Collectively the vertical momentum and circulation, which are contributors*

to lift, support that there is similar LEV formation for both linear and non-linear swept-back leading-edge geometries.”

“Table 1: Circulation for each angle of attack and spanwise measurement plane for the swift-style and delta wing geometries. Circulations are given in m^2s^{-1} .”

Angle of Attack	Spanwise Plane	Circulation: Swift	Circulation: Delta
0°	25%	0.00×10^{-3}	0.00×10^{-3}
	50%	0.00×10^{-3}	0.00×10^{-3}
	75%	0.00×10^{-3}	0.63×10^{-3}
20°	25%	4.23×10^{-3}	5.92×10^{-3}
	50%	6.45×10^{-3}	10.5×10^{-3}
	75%	4.72×10^{-3}	4.45×10^{-3}
30°	25%	10.9×10^{-3}	5.88×10^{-3}
	50%	8.73×10^{-3}	10.9×10^{-3}
	75%	7.85×10^{-3}	5.73×10^{-3}

L281-283: “Vertical momentum along the wing chord was also examined as an indicator of transport, and circulation was estimated for each wing geometry at each spanwise location and angle of attack.”

Minor comments

- Line 251: It is unclear how the vertical momentum distribution is calculated. What distance from the surface is the velocity sampled? Please clarify.

Thanks for pointing out the confusion. We have revised the explanation to include this information.

L244-250: “To unambiguously identify negative vertical velocity, we consider vertical displacements (from the PIV measurements) of at least 1.5 pixels in the downward direction as indicative of negative velocities (i.e., $v < -0.03 \text{ ms}^{-1}$). At each position along the chord, the mean of these negative vertical velocities over the region beneath the wing was computed and squared to estimate the vertical momentum distribution over the chord per unit span (ρv^2), where $\rho = 1000 \text{ kg m}^{-3}$, and is shown in Figure 7 for various planes and angles of attack.”

- Line 253: The velocity-squared momentum transfer term is not necessarily proportional to the total lift. It is directly proportional to the lift due to momentum transfer, but the total lift depends strongly on the surface pressure distribution. Information on the pressure distribution would be required to determine the contribution of momentum transfer to the total lift. It is recommended to limit the generalization to the total lift in the absence of pressure data.

Thanks for this comment. We have rephrased this sentence to avoid this confusion.

L250-252: “This distribution integrated over the chord is proportional to the vertical forces per unit span acting on the wing and is used to evaluate the similarity of the LEV as a mechanism contributing to (total) lift generation for the delta and swift wings.”

6. Figures 4, 5 and 6: The gradient in the colormap appears to give only three steps in the calculated vorticity contours. It is recommended to increase the resolution in the colormap to give a better picture of the variation in vorticity.

Thanks. We have adjusted Figures 4, 5, and 6 to include 10 contour levels.

“Figure 4: Mean vorticity contours for the quarter plane of the delta wing at $\alpha = 0^\circ$ (a), $\alpha = 20^\circ$ (b), and $\alpha = 30^\circ$ (c) and swift wing at $\alpha = 0^\circ$ (d), $\alpha = 20^\circ$ (e), and $\alpha = 30^\circ$ (f).”

“Figure 5: Mean vorticity contours for a delta wing at $\alpha = 20^\circ$ with contours of mean swirling strength (a,b,c) and mean Q criterion (d,e,f) overlaid. Each column depicts a different spanwise plane with the first column showing results for the quarter plane (a,d), the second column showing half plane results (b,e), and the third column showing results for the three-quarter plane (c,f).”

“Figure 6: Mean vorticity contours for a swift wing at $\alpha = 20^\circ$ with contours of mean swirling strength (a,b,c) and mean Q criterion (d,e,f) overlaid. Each column depicts a different spanwise plane with the first column showing results for the quarter plane (a,d), the second column showing half plane results (b,e), and the third column showing results for the three-quarter plane (c,f).”

7. Reference 4: Palhamus should be Polhamus

Thank you for bringing this to our attention. The necessary change has been made.

References:

“4. Polhamus EC. 1966. *A Concept of the Vortex Lift of Sharp-edge Delta Wings Based on a Leading-edge Suction Analogy*. U.S. National Aeronautics and Space Administration. Langley Research Center, Hampton, VA”

Response to Reviewer 2

Our point-by-point responses are included below where manuscript changes are denoted in italic and are referred to by line number of the revised manuscript.

Your effort is undermined somewhat by a general understanding that swift wings and an array of wing shapes from other animals, all of which have curved leading edges, all generate LEV's. It would more substantially advance our understanding of the effects of leading edge geometry upon LEV formation if your experimental design would have tested a broader range of variation in curvature until significant differences were observed between curved and straight leading edges. That written, your methods and analysis appear robust for the specific case you tested.

Thank you for reviewing the paper and for the constructive comments. We have chosen the swift wing as a case study because unlike other birds that generate LEV during flapping, the swift generate LEV during gliding, too, which is an uncommon feature amongst birds (Videler et al., 2004). The choice of comparison to a delta wing is because this is a classical wing that is known to generate an LEV and has been utilized widely in the aerospace industry. We have clarified this rationale in the revised manuscript in consideration of your comments, with specific changes identified below.

Specific Comments

1. I encourage you to recast your presentation more clearly as a test of a hypothesis than as a descriptive comparison. This applies to the abstract (lines 16-17) and later in the introduction and conclusions.

Thanks for your comment. We have revised our presentation, including the following changes:

Line 16-21: *“We hypothesize that non-linear swept-back wings generate a vortex in the leading-edge region, which can augment the lift in a similar manner to linear swept-back wings (i.e., delta wing) during gliding flight. Particle image velocimetry experiments were performed in a water flume to compare flow over two wing geometries: one with a non-linear sweep (swift-like wing) and one with a linear sweep (delta wing).”*

Line 88-89: *“The present study is an investigation of LEV formation comparing vortex development over a non-linear swept-back wing (swift) to that over a linear swept-back wing (delta).”*

Line 297-299: *“In all, these results reveal no major distinctions between the linear and non-linear swept-back wings on LEV formation, indicating that non-linear swept-back wings in addition to linear swept-back wings are capable of forming and maintaining a steady LEV.”*

2. Lines 22-23 (and conclusions). Your major conclusion is relatively weak, as we already understand that a variety of animal wings can generate LEV's.

While animal wings (insects, birds, bats) commonly generate LEV during flapping flight this is not the case during gliding flight; the case study herein. During gliding, the most well-known case to exhibit an LEV is the swift (Videler et al., 2004), and there exists speculation on LEV generation in owls during gliding as well (Kroger et al., 1972). Therefore, the conclusion provided herein is more generic than just the swift as many birds have a non-linear leading edge of their wing planform. We have revised the text you referenced to better reflect our findings.

L24-25: *“These similarities suggest that sweep geometries other than a linear sweep (i.e., delta wing) are capable of creating LEVs during gliding flight.”*

3. Line 31: Delete either “aerodynamic forces such as” or “lift and drag” as what other aerodynamic forces are there under consideration?

We have revised the sentence.

Line 33-34: *“Aerodynamic forces play a major role for a given task with respect to flight performance.”*

4. Line 34: LEV's generate greater pressure differentials on the wings and whether this is considered lift or drag is depended upon the angle of attack of the wing. My point is that LEV's are not a “free ride” in that they increase lift but also drag, and this inherent cost is worth including in the introduction.

We agree with the reviewer's comment. We have added a sentence mentioning the drag contribution of an LEV, and also added a reference regarding this point for the reader's information.

Line 45-46: *“However, this lift comes at a cost and it is noteworthy that LEVs also increase drag at all non-zero angles of attack⁶.”*

Reference

“6. Sane, Sanjay P. “The Aerodynamics of Insect Flight.” Journal of Experimental Biology, vol. 206, no. 23, Dec. 2003, pp. 4191–208. jeb.biologists.org, doi:10.1242/jeb.00663.”

5. Line 64: This sentence should be recast in terms of advance ratio. When the wings are moving faster than the forward speed of the animal, unsteady aerodynamic effects may dominate. The flapping motions may be just as steady (or unsteady) in slow flight as during cruising flight.

We apologize for the confusion. We agree with the reviewer that LEV formation over flapping animal wings depends on the coupling between the flapping frequency and forward speed as well as other conditions. However, we focus in this paper on gliding mode where no flapping is considered. Therefore, to avoid confusion, we have removed this sentence.

6. Line 78: Define “strong” in quantitative terms.

Thanks. This word should not have been there – we have removed it from the sentence.

Line 78-80: *“Swift wings are characterized by a non-linear sweep-back angle, which differs from delta wings that have a linear sweep-back angle.”*

7. Results: The beginning of the results (lines 166-179) is a restatement of the introduction and methods. This should be deleted as it is redundant. Several subsections of the results begin with a restatement of the introduction or methods (or in a couple of cases, conclusions). This includes lines 189-193, 202-207, and 246-255. These cases should be deleted as they are redundant, or components of them should be moved to the introduction or methods.

We appreciate the reviewer’s comment and have edited the noted sections to remove repetition. Several lines were removed including 166-172 and 202-203. We have verified that other information presented in these lines is the only place that information appears in the manuscript. Thanks.

8. Figure 1: Consider adding lines to indicate the span-wise locations at which you sampled flow using PIV

Thanks for this suggestion. We have amended it.

“Figure 1: 3D CAD renderings of the delta (a) and swift-like (b) wings. These wing models were created using a Mark Forged 3D printer with onyx fiber. Black lines on wings show the PIV cross sections measured at 25%, 50%, and 75% of the span.”

9. Figure 2: This figure isn't needed, as your setup is standard.

We agree that the PIV setup is standard, but the inverted nature of the wing is most easily visualized with the inclusion of this figure. We therefore think the figure provides additional clarification of the experimental setup.

10. Figures 7 and 8: Consider combining these, using two different line colors or solid versus dashed. This would facilitate comparison of the two sets of results.

Thank you for this comment. We have combined the two figures into one and adjusted the manuscript as necessary to reflect this change. The revised Figure 7 is included below.

“Figure 7: Vertical momentum distribution over the chord per unit span for the delta wing (black) and for the swift-based wing (red) for each angle of attack (rows) and each plane (columns). a, b, and c depict results for $\alpha = 0^\circ$, d, e, and f show results for $\alpha = 20^\circ$, and g, h, and i show results for $\alpha = 30^\circ$. Results for the quarter plane is shown in a, d, and g, for the half plane in b, e, and h, and for the three-quarter plane in c, f, and i. The respective vertical lines denote the leading and trailing edge of each wing in each plane.”